# Relationship between mobility and road traffic injuries during COVID-19 pandemic— The role of attendant factors

Kandaswamy Paramasivan[1,2]*, Rahul Subburaj[3,4], Venkatesh Mohan Sharma[4], Nandan Sudarsanam[2,4]

1 General of Police/Vigilance and Anti-corruption, Government of Tamil Nadu, Tamil Nadu, India, 2 Department of Management Studies, Indian Institute of Technology, Madras, India, 3 Department of Civil Engineering, Indian Institute of Technology, Madras, India, 4 Robert Bosch Centre for Data Science and AI, Indian Institute of Technology, Madras, India

* kandy@berkeley.edu

**Data Availability Statement:** All relevant data are within the paper and its Supporting Information files.

**Funding:** The authors acknowledge the support provided by the Robert Bosch Centre for Data

## Abstract

This study investigates the important role of attendant factors, such as road traffic victims' access to trauma centres, the robustness of health infrastructure, and the responsiveness of police and emergency services in the incidence of Road Traffic Injuries (RTI) during the pandemic-induced COVID-19 lockdowns. The differential effects of the first and second waves of the pandemic concerning perceived health risk and legal restrictions provide us with a natural experiment that helps us differentiate between the impact of attendant factors and the standard relationship between mobility and Road Traffic Injuries. The authors use the auto-regressive recurrent neural network method on two population levels–Tamil Nadu (TN), a predominantly rural state, and Chennai, the most significant metropolitan city of the state, to draw causal inference through counterfactual predictions on daily counts of road traffic deaths and Road Traffic Injuries. During the first wave of the pandemic, which was less severe than the second wave, the traffic flow was correlated to Road Traffic Death/ Road Traffic Injury. In the second wave's partial and post lockdown phases, an unprecedented fall of over 70% in Road Traffic Injury—Grievous as against Road Traffic Injury— Minor was recorded. Attendant factors, such as the ability of the victim to approach relief centres, the capability of health and other allied infrastructures, transportation and medical treatment of road traffic crash victims, and minimal access to other emergency services, including police, assumed greater significance than overall traffic flow in the incidence of Road Traffic Injury in the more severe second wave. These findings highlight the significant role these attendant factors play in producing the discrepancy between the actual road traffic incident rate and the officially registered rate. Thus, our study enables practitioners to observe the mobility-adjusted actual incidence rate devoid of factors related to reporting and registration of accidents.

Science and Artificial Intelligence (RBCDSAI),
Indian Institute of Technology Madras, India
(SB20210605MSRBCX008658). The funders had
no role in study design, data collection and
analysis, decision to publish, or preparation of the
manuscript.

**Competing interests:** The authors have declared
that no competing interests exist.

## Introduction

### Background

From its onset in early 2020 till about the middle of 2021, coronavirus infections across most
countries displayed two or three wave patterns in terms of reported cases and related fatalities.
To contain the spread of the virus, various governments worldwide had from time to time
enforced specific orders with varying degrees of restrictions depending on a host of conditions,
which often took into consideration the local circumstances. The first wave of COVID -19 was
more intense and severe than the second in most western economies [1–4]. However, in India,
compared to the first wave, the second wave of COVID-19 had a far more devastating effect in
terms of exponentially spiralling cases, overwhelming health infrastructure, lacking essential
medical supplies, and an increasing number of fatalities among the younger population.
Despite the largely deleterious effects of lockdown due to COVID-19 worldwide, there were
certain unintended tangential positive effects. Among these positive externalities, reduced air,
sound and water pollution, and reduced traffic crashes have been extensively noted [5].

### Literature on road traffic accidents during regular times and the pandemic

Several research papers have documented the correlation between traffic volume and crash
rates. The meta-analysis study by Hoye et al. [6] reveals that traffic crashes increase with an
increase in traffic volume. Retallack et al. [7] report that at lower traffic volumes, a linear rela-
tionship exists between traffic volume and crash frequency, and at higher traffic volumes, a
quadratic relationship exists between the two. Ahangari et al. [8] show that the two factors of
vehicle miles travelled and vehicles per capita have the strongest impact on traffic fatality rates,
and a study by Segui-gomez et al. [9] reveals that even a small reduction in the number of kilo-
metres travelled, contributes significantly to protection from crashes and resulting injuries.
The above studies were during regular or non-Pandemic times.

   For this paper, the authors have thoroughly studied the research on traffic safety in the con-
text of the pandemic-induced lockdown. A majority of the total of 51 papers in the category
report a drop in traffic fatalities and injuries during the pandemic, while only half a dozen doc-
ument increased road crashes and fatalities on account of pandemic-induced lockdowns.
Research studies [10] covering 11 countries for a period of six months of pandemic-related
restrictions suggest that stringent lockdown measures and high residential mobility substan-
tially reduced the crash fatality ratio (CFR). For example, most sub-groups of the population
in West Virginia altered their travel behaviours, resulting in lower motor-vehicle injury rates
[11]. Similar observations were made in other places across the world where restricted mobility
resulted in a marked reduction in the areas of traffic violations, the number of people involved
in crashes, and deaths. A study [12] of the effects of COVID-19 pandemic-induced lockdown
in four Indian states reveals that traffic accidents decreased by 75.2%. Another study [13] finds
a significant decrease in the number of trauma victims from road traffic crashes (RTC) during
the complete lockdown (CL) in India, especially in the categories of influence of alcohol and
head injuries, fractures, and dislocations. While a vast majority of the research findings indi-
cate a reduction in accidents, there are instances of escalation in RTC, Road Traffic Injuries
(RTI), and Road Traffic Deaths (RTD) in a few places, such as the USA [14–16] and the state
of Connecticut within it [17].

### Present research–A natural experimental study

While there are established research findings that show that mobility is proportional to RTC,
other attendant factors such as the ability of the victims to access relief centres, hospitals and

other legal facilities had not been studied earlier. This is because attendant factors remain more or less constant during normal times for a given location. However, during the pandemic, these services were thoroughly overwhelmed, and resources were stretched, due to which the access of road traffic victims to these services was severely restricted or drastically changed. The pandemic thus provides a timely opening to study the effect of these factors as a natural experiment.

So far, research studies have primarily assessed the impact of the first and second waves of the pandemic on traffic safety. To the best of the authors' knowledge, a comparative study of the impact of pandemic-induced lockdown on RTC/RTI/RTD in the first and second waves has not been undertaken. The two waves of the COVID-19 pandemic provide an opening for this study to have been conducted as a natural experiment, where two separate factors—mobility and a set of attendant factors, such as levels of access road traffic victims have to trauma centres, the robustness of health infrastructure, and the response of police and emergency ambulance services—have significant influence over instances of RTD/RTI. The methods used in various studies conducted so far have either been descriptive statistical analysis techniques, such as Difference in Difference, Regression-Discontinuity Design, Mixed Regression Model, or using Auto-Regressive Integrated Moving Average (ARIMA) models [18–20]. In this study, the authors estimate the causal inference on traffic safety on account of the COVID-19 lockdowns using an autoregressive recurrent neural network-based forecasting model for counterfactual prediction in the context of an altered number of road crashes, resulting in fatalities and different injury outcomes for the state of Tamil Nadu (TN). Recurrent neural networks, a relatively new but sophisticated method with high forecasting accuracy, have been used extensively in the fields of natural language processing and supply chain management. In this study, recurrent neural networks have been used to forecast RTD/RTI.

As the pandemic has stretched and overwhelmed the law-organizing and caregiving institutions of our societies, it presents to us an ideal opening to study the critical role of attendant factors in challenging the well-established relationship between traffic flow and RTD/RTI in the second wave of the Pandemic in India. By the phrase 'attendant factors', the authors in the paper refer to the levels of access a victim has to first responders, which include the police, ambulance services, trauma centres, inclination of the victims to lodge complaints, and the robustness of health infrastructure. The attendant factors have a greater influence on the incidence of RTI-Grievous, where victims require hospitalization and immediate transportation to trauma centres, than the incidence of RTD and RTI-Minor, which are affected to a much lesser degree. Hence, the authors compare RTD/RTI-Minor to RTI-Grievous to corroborate the role of the attendant factors. Moreover, the authors compare Chennai, the largest metropolitan city in the state of TN, with TN as a whole and find that the incidence of RTI-Grievous is inherently different in the two regions, as citizens in Chennai have better access to hospitals than the dwellers of the rural regions of TN. The city also has a higher concentration of police personnel.

## Materials and methods

### Data

Road Traffic Crashes are registered as crimes in First Information Reports (FIRs) in the 1356 police stations in Tamil Nadu. Chapter XVI (sections 279 to 304) of the Indian Penal Code (IPC) deals with offences related to rash and negligent acts. This study focuses specifically on more commonly reported crashes in this category, namely, Road Traffic Injury–Minor, Road Traffic Injury–Grievous, and Road Traffic Deaths.

The movement restrictions issued by the Government during 2020 and 2021 can be classified into two categories. During the period of complete lockdown (CL-2020), the restrictions

were severe, heavily impairing the mobility of people and vehicles. However, during partial lockdowns (P.L.s), there were partial and periodic relaxations in mobility for specific categories of people, goods, and services, and the opening hours of business establishments. Restrictive orders differed during the two waves. In the first wave, CL-2020 was implemented on March 23, 2020, and lasted till April 30 2020. Relaxations were introduced from May 1 2020, and the period from then to June 8 2020, is considered PL-2020. Most restrictions were removed on September 1 2020, after the post-lockdown period (Post-L-2020) began. In 2021, during the second wave of the pandemic, the Government initially introduced a partial lockdown (PL-One-2021), which commenced on April 10, 2021, and lasted till May 5, 2021. This was immediately followed by CL-2021 which witnessed severe restrictions from May 6 2021, to June 7, 2021. The restrictions were subsequently relaxed, and the second partial lockdown (PL-Two-2021) was introduced from June 8 to July 6 2021. Thus, we can see that during the second wave, CL-2021 was sandwiched between PL-One-2021 and PL-Two-2021. Most of the curbs were removed effective September 1, 2021 and the period September 1, 2021 to October 8, 2021 is treated as post lockdown in 2021 (Post-L-2021).

**Different phases in Wave One of COVID-19 in 2020**
Training Data and Model Building: January 1, 2010, to December 31, 2019
Validation Data: January 1, 2020, to March 22, 2020
CL-2020: March 23, 2020, to April 30, 2020
PL-2020: May 1, 2020, to June 8, 2020
Post-L-2020: September 1, 2020, to October 8, 2020
**Different phases in Wave Two of COVID-19 in 2021**
PL-One-2021: April 10, 2021, to May 5, 2021
CL-2021: May 6, 2021, to June 7, 2021
PL-Two-2021: June 8, 2021, to July 6, 2021
Post-L-2021: September 1, 2021, to September 30, 2021

## Method

Forecasting crime with a high degree of accuracy is a great challenge, given the complex nature of the phenomenon of the occurrence of crime. It depends on many criminogenic factors and the correct recording of crime. Traditional models used for forecasting crime provide only point estimates for the future values of the time series. In contrast, a model which supplies a range or confidence interval for the predictions indicates the reliability of the estimates and also helps the police in planning and deploying strategic preventive actions by effective mobilization of resources. In the case of traditional models, manual specification of functional form and deciding the frequency at which the input time series must be aggregated to get the best predictions further complicates and eventually deteriorates the forecast quality. To overcome these drawbacks, this study employs an autoregressive recurrent neural network-based model called DeepAR [21] to forecast crime. The model creates its features by aggregating the input time series with different frequencies, thereby learning any functional form without any intervention from the modeller. Further, the model does not assume that the error term is normally distributed (which is the case in traditional models like ARIMA); rather, the error term is accommodated in a wide array of both continuous and discrete likelihood functions such as the negative binomial, which are more appropriate for non-negative count data. Moreover, the DeepAR model is highly suitable for learning from large time series data compared to commonly used models like ARIMA and exponential smoothing etc. This ability of the DeepAR model to capture short and long-term dependencies in a large time series is attributed to the fact that they have a huge number of parameters which need more data to train thereby

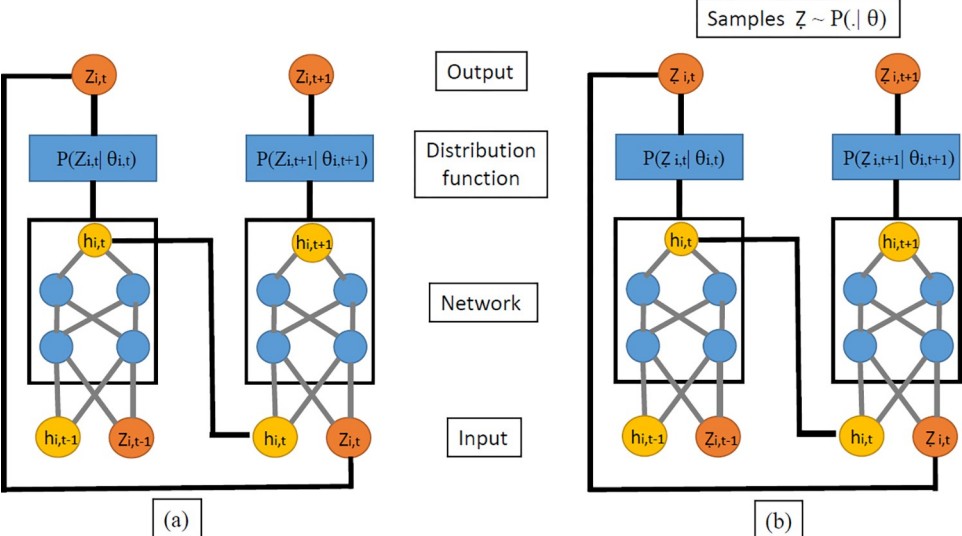

**Fig 1.** A schematic representation of the DeepAR model during (a) Training phase (b) Testing phase.

naturally making them suitable for big data. This is relevant in this study as the data spans across a decade and is also from two geographic locations making it complex.

**Auto-regressive recurrent neural network—DeepAR.** Recurrent Neural Network (RNN) is a class of Artificial Neural Networks that are designed to model sequential data. In sequential data, the occurrence of a particular value in the sequence depends on its previous values. RNNs accommodate this dependence between the inputs by passing on the value of the sequence in the previous time step to the current time step, making them ideal for time series data which is sequential in nature (Fig 1).

The input time series is divided into two distinct periods - The first period is the training or conditioning period which corresponds to time $t = 0,1,2,3,4,\ldots,t0-1$, and the latter is the prediction range which is from $t_0, t_0+1, t_0+2,\ldots,T$, where $t_0$ and $T$ represent the first and last steps of the forecasting period. In the training range (Fig 1), at every step t, the input to the network is the value of the time series at the previous time step $z_{i,\,t-1}$ and the network output at the previous time step $h_{i,t-1}$. The output of the recurrent neural network at time step t for time series i is given by

$$h_{i,t} = h(h_{i,t-1}, z_{i,t-1}, \Theta) \tag{1}$$

In order to generate a probabilistic forecast, a likelihood function specified by the modeller is used. The output of the recurrent neural network is used to compute the parameters of the likelihood function $\theta_{i,\,t}$ using the equation given below

$$\theta_{i,t} = \theta(h_{i,t}, \Theta) \tag{2}$$

Where $\Theta$ represents the parameters of the recurrent neural network.

**Training.** The parameters of RNN h(.) and θ(.) can be learned by maximizing the log-likelihood function given below:

$$\sum_{i=1}^{N}\sum_{t=t_0}^{T} \log p\ (z_{i,t}|\theta(h_{i,t})) \tag{3}$$

At each time step t, the target value at the previous time step $z_{t-1}$ as well as the previous network output $h_{t-1}$ is used to compute the $h_t$. This $h_t$ is then used to calculate parameters (μ and

σ in case of Normal distribution or α and μ in case of binomial) for the probability of the likelihood function, which is used for training the recurrent neural network parameters. At time t = 1 or the first step, the parameters are initialized to zero.

Any likelihood function measures the fit of the statistical model to a sample of data for given values of the unknown parameters. In this study, Negative binomial distribution with parameters mean (μ) and the shape parameter (α) is used for modelling the time-series data of crimes as they are non-negative integers. The output from the last layer is subjected to an affine transformation followed by a softplus activation to ensure a positive value of the mean and shape parameter of the likelihood function.

**Prediction.**　Upon training the model up to the t time step, it then comes to predicting the value of the time series during the prediction range. Forecasting for a point $z_{t+1}$ involves a four-step process. First, the values of the time series and the network prediction at the specified lags are taken. Second, these values form the input layer of the RNN architecture and with the parameters learned in the training phase the output of the network at time step t+1 is calculated. Third, this output goes as the input to the likelihood function which along with the parameters are used to calculate the mean and shape parameter of the negative binomial distribution. The fourth step involves drawing n samples from the distribution whose median value is given as the output or the forecast at time step t+1. The four-step process is summarized in Eq (4).

$$h_{t+1} = f(h_t, Z'_t; \ \Theta) - \text{Output of RNN}$$
$$Z_{t+1} = Median(Sample_{n \ times} l(h_{t+1}, ; \ \mu_{t+1}, \alpha_{t+1,})) \tag{4}$$

Where f and l represent the RNN and the likelihood function respectively. $Sample_{n \ times}$ represents the process of drawing n samples from the likelihood function. In this study 100 samples are drawn. This process is repeated till time step T is attained, which marks the ends of the prediction range.

Next is the confidence interval estimation for each time step in the prediction range. The n samples drawn from the distribution are used to calculate the quantiles needed for the estimation of the confidence interval. For example, to estimate the 95% confidence interval (used in the study), 0.025 and 0.975 quantiles are calculated, and the interval formed by them represents the required confidence interval.

Now that we have the predicted and actual time series, we can use them to quantify the lockdown effect on crime trends. The differences observed between the predicted and actual time series are based on the lockdown imposed by the Government. Since we know the exact start and end of each such intervention period, we can quantify the effect of lockdown. As the distributions were non-normal, the Wilcoxon sign rank test is employed to conclude whether the difference in means of predicted and actual values of a crime during a period is statistically significant. The effect size also is computed to quantify the effect of lockdown on crimes instead of inferring if lockdown has had an effect on crimes. Cliff's delta measure is used for quantifying the effect since Cohen's d (typically used measure) is calculated only using the means and standard deviations of the two groups being compared (as are other common effect sizes, such as Glass's delta or Hedges' g), which makes it inappropriate for the quantification of the effect in the absence of normality of data [22]. On the other hand, Cliff's delta is a robust measure that can be used in cases where the parametric assumptions are not met [23].

In order to evaluate the forecast accuracy of the DeepAR model with the most popular forecast techniques such as Auto-regressive Integrated Moving Average (ARIMA), Holt-Winters, and state of the art models like Generalized Additive Model (GAM), Bayesian Structural Time Series (BSTS) model, an appropriate error metric is needed. This study employs Weighted

Mean Absolute Percentage Error (WMAPE) to check the prediction accuracy of 19 different types of crime series as it is unit-free and easily interpretable.

**Experiments.** The dataset for the state of Tamil Nadu used in the study ranges from January 1, 2010 to June 30 2021 with day-wise frequency. The model was trained with data from January 1, 2010 to December 31, 2019 and validated with data from January 1, 2020, to February 15, 2020. The testing period was from February 16, 2020 to June 30, 2021. Several hyperparameters that define the RNN model structure need to be specified before training the model. Hyperparameters like the number of hidden layers and the number of cells in each layer were tuned to jointly achieve a minimum loss value in the train and validation set.

For district-level data, the weekly frequency was used instead of a daily frequency to avoid non-zero values in the time series, making it difficult for the model to fit the training data and subsequently leading to poor forecasts.

Following are the hyperparameters (with their range) of the DeepAR model tuned in the study.

Likelihood function = Negative Binomial

Batch size = 16

Epochs = 1 to 300

Number of hidden layers = 3

Number of cells in each hidden layer = 80

Patience = 10 to 40

Decay factor = 0.2 to 0.9

Learning rate = 1e-3 to 1e-20

The learning rate parameter can be adjusted to avoid under-fitting or over-fitting of the time series by controlling the training process. Patience parameter indicates the number of steps to wait before early stopping and the decay factor controls the reduction in learning rate parameter. This model is implemented using the GluonTS package in python and exact code used in this study can be found in our Github repository (link).

## Results

### Changes in mobility in all spheres of activity

In this section, the authors first discuss how the second wave of COVID-19 in 2021 was different from the first wave in 2020 in terms of severity, infection rate, spread and fatality rate. In order to primarily contain the spread of the pandemic during both the waves, the Government had enforced stay-at-home orders in different phases with varying degrees of relaxations commensurate to the magnitude of the pandemic crisis. The mean/standard deviation/maximum of the number of infections during the first and second waves were 2403/2223/6993 and 12265/11626/36184, respectively. Similarly, the fatality rate per day were 34/36/127 and 153/163/483, respectively. The below graph (Fig 2) illustrates how the mean infection and fatality in the second wave were respectively almost six and five times more severe than those during the first wave. Due to the overwhelming circumstances of the second wave, the hospital and other health infrastructure faced a deluge of infected people.

The restrictive orders of the Government impacted the mobility of people in different activity zones, and this impact on mobility had been along the expected lines. The authors have used the Google Community Mobility database [24] for the state of Tamil Nadu and Chennai City for the relevant time period. The Google Community Mobility Report provides the changes in mobility in six sectors. There are six spatial domains where the human mobility trends over a period of several months are available and they are Retail and Recreation, Transit Stations, Workplace, Groceries and Pharmacy, Parks and Residential. In all the areas, the data

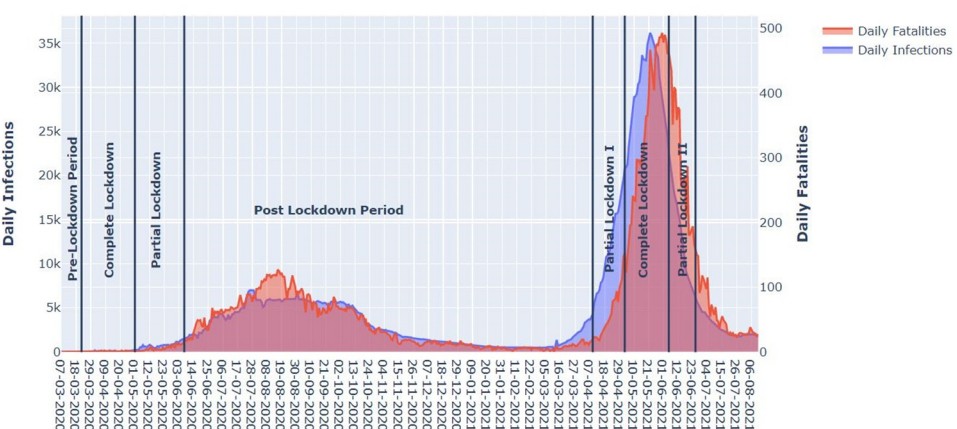

**Fig 2. Graph of daily number of infections (Y-axis–Left in thousands) and fatalities (Y-axis-Right) during two waves of pandemic (X-axis day timeline).**

measures a change in the total number of visitors except the last category of residential sector which shows a change in the duration of time. Both compared to baseline days. The baseline day is the median value from the five-week period. As we are interested in the comparative analysis of mobility in different periods, having the absolute numbers for mobility is not important. As expected, there was a considerable increase in percentage change from the baseline value in mobility for residential neighbourhoods as stay-at-home orders ensured people remained indoors. There was a steep decline in percentage change in mobility from the baseline in all the other spheres of activities, such as Retail and Recreation, Grocery and Pharmacy, Parks, Transit Stations, and Workplaces (Fig 3). Importantly, the quantum of such reduction was more in respect to complete lockdown and partial lockdown in the first wave than in the second wave as shown in Tables 1 and 2.

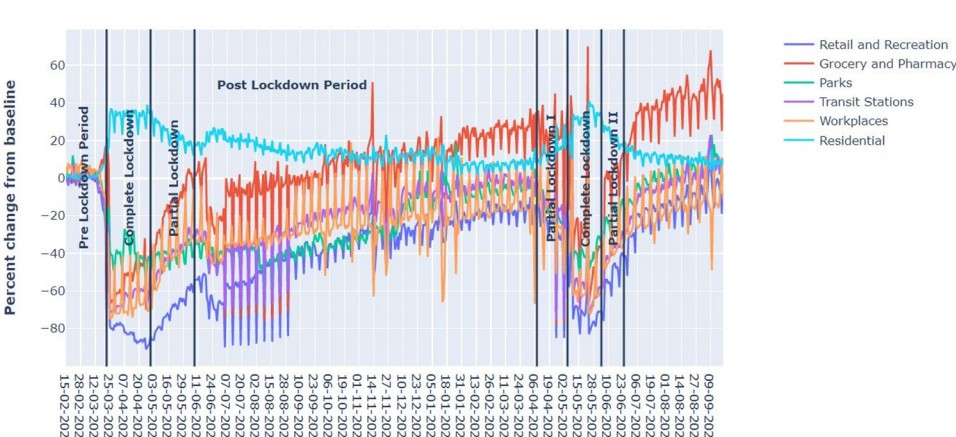

**Fig 3. Graph of percentage change in mobility from baselines in six spatial domains during the first and second waves of the pandemic in 2020 and 2021.** Source: Google mobility index.

**Table 1. Descriptive statistics of the changes in mobility (in %) from baseline during various lockdown phases in both waves in Tamil Nadu in the six spheres of activity.**

| Period/Zone | | Retail and Recreation | Grocery and Pharmacy | Parks | Transit Stations | Workplaces | Residential |
|---|---|---|---|---|---|---|---|
| CL 2020 | Mean | -80.4 | -52.21 | -39.38 | -62.03 | -64.33 | 32.10 |
| | Std | 11.85 | 17.71 | 8.79 | 9.30 | 11.70 | 5.72 |
| | Max | -26 | 21 | -15 | -24 | -29 | 39 |
| PL 2020 | Mean | -70.21 | -16 | -40.82 | -41.56 | -39.21 | 22.77 |
| | Std | 8.87 | 14.17 | 4.7 | 8.91 | 14.31 | 5.49 |
| | Max | -54 | 5 | -32 | -26 | -6 | 36 |
| Post-L 2020 | Mean | -39.65 | -0.57 | -36.8 | -22.7 | -24.13 | 13.3 |
| | Std | 4.8 | 7.8 | 3.68 | 3.7 | 11.7 | 2.8 |
| | Max | -34 | 18 | -29 | -16 | 3 | 19 |
| PL–One 2021 | Mean | -47 | -4.73 | -34.04 | -33.08 | -36.19 | 19.42 |
| | Std | 11.57 | 16.15 | 9.17 | 10.88 | 9.94 | 3.79 |
| | Max | -25 | 27 | -15 | -13 | -11 | 26 |
| CL 2021 | Mean | -60.42 | -22.15 | -40.03 | -47.67 | -44.61 | 23.55 |
| | Std | 5.96 | 8.87 | 6.39 | 5.27 | 8.56 | 2.8 |
| | Max | -44 | 1 | -25 | -35 | -25 | 28 |
| PL–Two 2021 | Mean | -35.07 | 9.31 | -19.66 | -23.97 | -27.48 | 13.93 |
| | Std | 7.93 | 10.46 | 5.27 | 6.52 | 5.33 | -15 |
| | Max | -20 | 28 | -9 | -12 | -15 | 19 |
| Post-L 2021 | Mean | -6.8 | 45.6 | 9.16 | 4.2 | -8.23 | 8.7 |
| | Std | 6.4 | 9.2 | 5.7 | 5.6 | 12.12 | 2.3 |
| | Max | 8 | 68 | 23 | 23 | 16 | 15 |

## Forecast accuracy of model

The hyperparameter tuning provides the best model. The lags included in the model are 1, 2, 3, 4, 5, 6, 7, 8, 13, 14, 15, 20, 21, 22, 27, 28, 29, 30, 31, 56, 84, 363, 364, 365, 727, 728, 729, 1091, 1092, 1093. These lags can be segmented into four categories namely short term dependencies (last week's values), weekly-, monthly-, and yearly seasonality. The autoregressive terms and the network output at these lags are used for predictions.

The optimal value of the hyperparameters is given below

Learning rate = 1e-15

Patience = 20

Decay factor = 0.6

Epochs = 250

To validate the model, the forecast accuracy of all the models were tested with the validation data, using WMAPE as the appropriate error metric as shown in Table 3.

## Impact of lockdown on RTD/RTI

**Road Traffic Deaths (RTD).** Among the different outcomes of RTC, RTD is accorded with higher priority than RTI as it concerns loss of life. In line with the mobility trend, the drop in RTD when compared with the counterfactual has been the most substantial during the complete lockdowns in wave one and wave two, with drops measured at 78.73% (cliff delta of -0.90) and 64.45% (cliff delta of -1), respectively. Though the drops for the periods of PLs during both waves were significant, they were much less than the ones recorded during the CLs. The declining percentage during CL-2020 was 78.73% (cliff delta of -1) as opposed to the

**Table 2. Descriptive statistics of the changes in mobility (in %) from baseline during various lockdown phases in both waves in Chennai in the six spheres of activity.**

| Period/Zone | | Retail and Recreation | Grocery and Pharmacy | Parks | Transit Stations | Workplaces | Residential |
|---|---|---|---|---|---|---|---|
| CL 2020 | Mean | -88.87 | -61.94 | -88.48 | -85.48 | -78.6 | 36.38 |
| | Std | 8.52 | 19.38 | 8.52 | 5.8 | 9.48 | 6.3 |
| | Max | -46 | 18 | -56 | -58 | -48 | 45 |
| PL 2020 | Mean | -80.5 | -39.3 | -92.2 | -75.5 | -62.07 | 30.07 |
| | Std | 23.3 | 9.8 | 2.98 | 4.65 | 11.88 | 5.39 |
| | Max | -69 | -25 | -84 | -68 | -32 | 41 |
| Post-L 2020 | Mean | -51.7 | -16.7 | -68.7 | -52.7 | -45.9 | 17.01 |
| | Std | 5.87 | 9.8 | 5.55 | 4.9 | 9.64 | 3.16 |
| | Max | -46 | 1 | -55 | -44 | -24 | 23 |
| PL–One 2021 | Mean | -46.3 | -1.69 | -48.69 | -40.42 | -40.3 | 16.19 |
| | Std | 15.5 | 26.9 | 15.49 | 14.12 | 14.89 | 5.04 |
| | Max | -29 | 29 | -25 | -26 | -7 | 27 |
| CL 2021 | Mean | -79.3 | -43 | -76.18 | -72.7 | -66.8 | 31.03 |
| | Std | 8.24 | 24.3 | 6.4 | 11.6 | 17.3 | 7.55 |
| | Max | -55 | 23 | -60 | -40 | -11 | 40 |
| PL–Two 2021 | Mean | -54.82 | -4.3 | -55.9 | -49.4 | -49 | 19.37 |
| | Std | 12.09 | 14.6 | 9.76 | 12.14 | 11.1 | 5.24 |
| | Max | -33 | 19 | -37 | -26 | -22 | 29 |
| Post-L 2021 | Mean | -23.83 | 24.23 | -24.3 | -16.6 | -28 | 9.83 |
| | Std | 7.4 | 11.52 | 7.47 | 5.4 | 12.07 | 2.9 |
| | Max | -15 | 43 | -16 | -9 | -3 | 20 |

percentage of 44.83% (cliff delta of -0.904) in PL-2020. Following a similar trend, the percentage value was 64.45% (cliff delta of -1) in CL-2021 as opposed to the percentage value of 46.52% (cliff delta of -1) in PL-One in 2021 and 44.68% (cliff delta of -0.94) in PL-Two-2021 in TN as shown in Table 4. In general, one could say that the pattern of decline in RTD during CL and PL phases in wave one and wave two are very similar (Fig 4).

When the model predicted the unseen test data during the validation period (pre-lockdown period), the forecasted range of daily counts of RTD is almost the same as the actual incidence of the events. The notable difference is in the post lockdown phases. When the curbs were removed during Post-L-2020, there was no difference in the actual and predicted number of RTD, and the decline was only partial, at 8.55% (cliff delta of -0.165). However, after the

**Table 3. Validation error (WMAPE) of five models for the unseen data (January 1, 2020 to March 22, 2020).**

| RTC | ARIMA | GAM | BSTS | HOLT-WINTERS | DeepAR |
|---|---|---|---|---|---|
| | WMAPE | WMAPE | WMAPE | WMAPE | WMAPE |
| Road Traffic Death | 0.162 | 0.173 | 0.167 | 0.171 | 0.143 |
| Road Traffic injury (Grievous) | 0.123 | 0.121 | 0.130 | 0.185 | 0.119 |
| Road traffic injury (Minor) | 0.116 | 0.107 | 0.159 | 0.238 | 0.119 |

List of Abbreviations.

GAM—Generalized Additive Model.

BSTS—Bayesian Structural Time Series.

DeepAR—Autoregressive Recurrent Neural Network–Implemented through DeepAR.

**Table 4. Statistics of RTD/RTIs in all phases during both waves for TN.**

| RTC/PERIOD | | RTD | RTI—Grievous | RTI—Minor |
|---|---|---|---|---|
| **CL 2020** | Actual | 10.63 | 16.47 | 29.42 |
| | Predicted Mean | 49.98 | 101.51 | 148.06 |
| | Pre. Mean CI[a] | 43.5, 56.46 | 81.27, 121.75 | 130.82, 165.3 |
| | Percentage Change | -78.73 | -83.77 | -80.13 |
| | Cliff delta | -0.90 | -1 | -1 |
| **PL 2020** | Actual | 30.05 | 42.68 | 79.08 |
| | Predicted Mean | 54.47 | 98.26 | 156.08 |
| | Pre. Mean CI | 42.7, 66.23 | 83.2, 113.32 | 131.77, 180.38 |
| | Percentage Change | -44.83 | -56.56 | -49.33 |
| | Cliff delta | -0.904 | -1 | -1 |
| **Post–L 2020** | Actual | 45.58 | 70.13 | 142.32 |
| | Predicted Mean | 49.84 | 81.17 | 143.86 |
| | Pre. Mean CI | 39.34, 60.34 | 64.87, 97.46 | 122.13, 165.58 |
| | Percentage Change | -8.55 | -13.6 | -1.07 |
| | Cliff delta | -0.165 | -0.59 | -0.028 |
| **PL-One- 2021** | Actual | 30.95 | 46.20 | 107.90 |
| | Predicted Mean | 57.87 | 98.91 | 165.72 |
| | Pre. Mean CI | 42.94, 72.79 | 81.74, 116.08 | 146.1, 185.33 |
| | Percentage Change | -46.52 | -53.19 | -34.89 |
| | Cliff delta | -1 | -1 | -1 |
| **CL 2021** | Actual | 18.73 | 23.40 | 52.47 |
| | Predicted Mean | 52.72 | 109 | 167.02 |
| | Pre. Mean CI | 46.33, 59.12 | 89.96, 128.05 | 142.72, 191.32 |
| | Percentage Change | -64.45 | -78.53 | -68.58 |
| | Cliff delta | -1 | -1 | -1 |
| **PL-Two-2021** | Actual | 30.95 | 46.20 | 107.90 |
| | Predicted Mean | 55.95 | 106 | 159.63 |
| | Pre. Mean CI | 45.29, 66.6 | 84.66, 127.35 | 133.31, 185.94 |
| | Percentage Change | -44.68 | -56.41 | -32.41 |
| | Cliff delta | -0.94 | -1 | -0.84 |
| **Post L -2021** | Actual | 25.77 | 10.43 | 122.80 |
| | Predicted Mean | 51.61 | 81.76 | 147.57 |
| | Pre. Mean CI | 40.22, 62.99 | 65.81, 97.71 | 125.47, 169.66 |
| | Percentage Change | -50.07 | -87.24 | -16.79 |
| | Cliff delta | -0.985 | -1 | -0.5 |

[a] Pre. Mean CI represents confidence interval of predicted mean.

complete removal of restrictions in wave two, during Post-L-2021, there was a substantial fall by 50.07% (cliff delta of -0.985) as shown in Table 4.

**Road traffic injuries-grievous.** The closeness of the values of predicted and actual daily counts of RTI-Grievous before the lockdown is proof of the accuracy of the model. Right from the onset of enforcement of first lockdown orders, the actual incidence of RTI-Grievous has been consistently lower in differing magnitudes than the predicted incidence during various phases, such as CL, PL and Post-L in the first and second waves of the COVID-19 pandemic. Further, the sharp and moderate decline in cases witnessed during CL and PL, respectively during the first wave, did not replicate itself in the second wave (Fig 5). During the first wave,

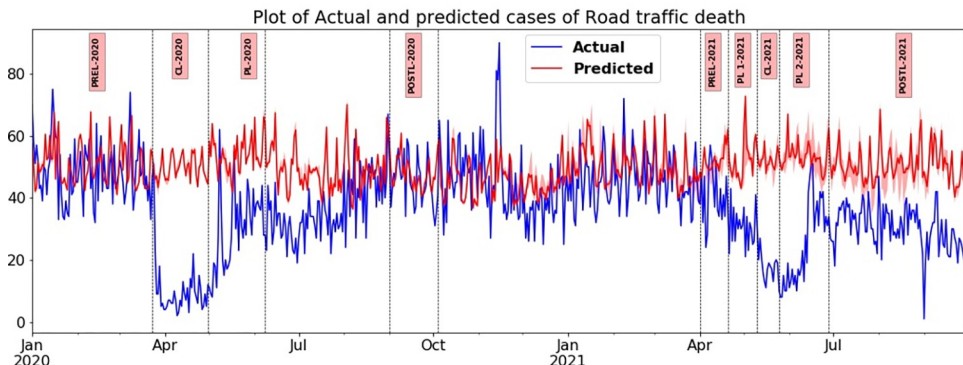

**Fig 4. Plot of daily number of road traffic deaths (RTD) in TN actual versus predicted during various phases of wave one and wave two in 2020 and 2021.**

there was a substantial fall as the percentage value decreased from 83.77% (cliff delta of -1) in CL-2020 to 56.56% (cliff delta of -1) in PL-2020. Similarly, in the second wave there was a drastic fall as the percentage values of decline were 78.53% (cliff delta of -1) in CL-2021, 53.19% (cliff delta of -1) in PL-One-2021 and 56.41% (cliff delta of -1) in PL-Two-2021 as shown in Table 4. In the second wave, relaxations of lockdown measures and increased mobility did not cause even a small upsurge, instead, the sharper decline in cases of the actual incidence of RTI-Grievous continued.

**Road Traffic Injuries-Minor.** This trend is quite similar to RTD in many respects (Fig 6). As noticed in RTD and RTI- Grievous in the CL phases of the first and second waves, RTI-Minor also plummeted by 80.13% (cliff delta of -1) and 68.58% (cliff delta of -1), respectively, when compared with the counterfactual values. During PL-2020 in the first wave, the cases of RTI-Minor were 49.33% (cliff delta of -1) less than the counterfactual. However, in PL-One-2021 and PL-Two-2021 during the second wave, the decline was 34.89% (cliff delta of -1) and 32.42% (cliff delta of -0.84), respectively.

The most distinguishing feature of RTI- Minor, when compared with other outcomes such as RTD and RTI-Grievous, is that the actual and predicted values were almost the same during the post lockdown phases in both the first and second waves in 2020 and 2021. The drop in the first wave was 1.07% (cliff delta of 0.028), and in the second wave, there was a moderate decline by 16.79% (cliff delta of -0.5); for details (Fig 6).

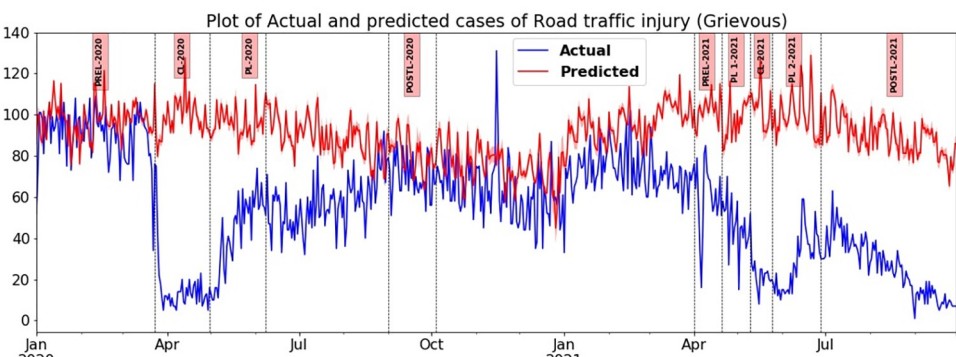

**Fig 5. Plot of the daily number of grievous Road Traffic Injuries (RTI-Grievous) in TN actual versus predicted during various phases of wave one and wave two in 2020 and 2021.**

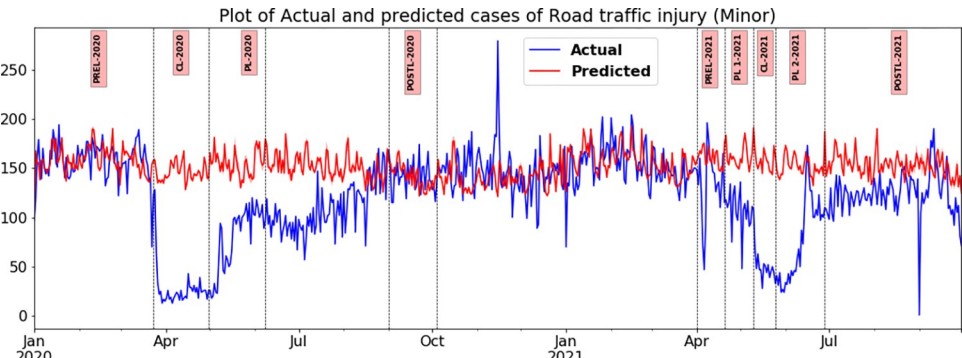

**Fig 6. Plot of the daily number of minor Road Traffic Injuries (RTI-Minor) in TN actual versus predicted during various phases of wave one and wave two in 2020 and 2021.**

## Discussion

The discussion on comparative analysis results is divided into four parts, each focusing on a particular frame of reference for comparing different categories of RTD/RTI during the various phases of lockdown in relation to mobility and other attendant factors in the course of the two COVID-19 waves in the state of Tamil Nadu and Chennai, its capital city The first part is the comparative analysis of the RTD/RTI between partial and complete lockdown within each wave, that is, CL-2020 vs PL-2020 and CL-2021 vs (PL-One-2021 and PL-Two-2021). The second part compares the RTD/RTI trend between the first and second waves of COVID-19 in Tamil Nadu, that is, CL-2020 vs CL-2021, PL-2020 vs PL-One-2021 and PL-Two-2021 and Post-L-2020 vs Post-L-2021. The third part elucidates the differential trends noticed in the varied outcomes of road traffic crashes, which are RTD, RTI-Grievous and, RTI-Minor. The fourth part compares the RTD/RTI trends noticed in Tamil Nadu and Chennai, largest metropolitan city of Tamil Nadu.

### Comparative analysis between complete and partial lockdowns within each wave

The first wave of COVID-19 in Tamil Nadu started in 2020—approximately five to six weeks after most parts of the US and Europe had reported the spread of the viral infections. This alerted the Government to the potential crisis and prompted the imposition of the strictest form of stay-at-home orders (CL) starting March 23, 2020. This was followed by PL when certain restrictions were lifted. The CL lasted 39 days, and the same number of days were considered for PL in the study. The relaxations paved for increased mobility, as noted in the Google Mobility Index for TN as shown in Table 1. The report shows the percentage change in mobility from baseline for six sectors: Retail and recreation (-80.46% vs -70.21%), grocery and pharmacy (-52.21% vs -16%), transit stations (-62.03% vs -41.56%) and workplaces (-64.33% vs -39.21%), along with the corresponding increase in mobility in residential areas (32.10% vs 22.77%). There is an unprecedented decline in RTD, RTI-Grievous, and RTI-Minor during CL-2020, measured at 78.73%, 83.77% and 80.13% as against the substantial fall at 44.83%, 56.56% and 49.33%, respectively during PL-2020 when pitted against the counterfactual prediction as shown in Table 4. In low- and middle-income countries, reversing the upward trends in RTC, RTD and RTI is considered to be a challenging task. Prior to the onset of COVID-19 pandemic, TN had managed to reverse the trends with a reduction of around 8%, an achievement hugely appreciated in the World Bank blogs [25]. In this context, a reduction in the magnitude of 70% during CL-2020 is certainly unprecedented. The fall in RTD/RTI in

**Table 5.  Effect size for various group comparisons among various phases of lockdown during first and second waves of COVID-19 in the state of Tamil Nadu and city of Chennai.**

| Cliff's Delta Effect Size | | PL-2020 vs CL-2020 | PLs-2021 vs CL-2021 | CL-2020 vs CL-2021 | PL-2020 vs PLs -2021 |
|---|---|---|---|---|---|
| **Tamil Nadu** | **Retail and Recreation** | 0.548, Large | 0.53, Large | -0.52, large | -0.73, large |
| | **Grocery and Pharmacy** | 0.47, medium | 0.51, Large | -0.35, medium | -0.85, large |
| | **Parks** | 0.06, negligible | 0.49, Large | -0.23, small | -0,84, large |
| | **Transit stations** | 0.47, medium | 0.47, large | -0.25, small | -0.35, medium |
| | **Workplaces** | 0.56, large | 0.54, large | -0.29, small | -0.35, medium |
| | **Residential** | -0.51, large | -0.5, large | 0.14, small | 0.44, medium |
| **Chennai** | **Retail and Recreation** | 0.53, large | 0.527, Large | -0.59, large | -0.69, Large |
| | **Grocery and Pharmacy** | 0.44, medium | -0.51, large | -0.35,medium | -0.79, large |
| | **Parks** | 0.13, negligible | 0.53, large | -0.67, large | -0.88, large |
| | **Transit stations** | 0.49, large | 0.50, large | -0.61, large | -0.76, large |
| | **Workplaces** | 0.55, large | 0.54, large | -0.43, large | -0.52, large |
| | **Residential** | -0.45, medium | -0.53, large | 0.38, medium | 0.65, large |

PL-2020 has been substantial but not as unprecedented as the values noted in CL-2020. This phenomenon is directly linked to the increased mobility contributing to higher traffic flows during PL-2020, as reported (as shown in Table 5) in the medium to a large effect size of different spheres of activity (Cliffs Delta 0.06, negligible to 0.56, large), when the distributions of mobility in CL -2020 are compared with PL-2020. The residential sector is the only sector that records the contrasting trend of increased mobility when restrictive orders are enforced as it is a natural consequence of lockdown when people are expected to remain indoors. Therefore, milder restrictive orders account for a lesser increase in mobility in this sector as against stricter orders. As in normal times, the attendant factors did not impact RTI.

In wave two, PL-One-2021 was followed by the consecutive periods, CL-2021 and PL-Two-2021, and then Post-L-2021. The decrease in mobility was less in wave two than in wave one. In the latter period, there was an absolute correlation between mobility and RTD/RTI. Furthermore, the mobility had a steep fall during CL and a moderate fall in PLs in the second wave, which was similar to the first wave's trend. For instance, from Table 1, the percentage decline in mobility from the baseline during PL-One-2021, CL-2021, and PL-Two-2021 in the second wave were 47.04%, 60.42%, 35.07%, respectively in the retail and recreation segment, and the corresponding fall in percentages in workplaces were 36.19%, 44.61%, 27.48% respectively. In fact, in the groceries and pharmacy sector, relaxations in restrictions saw a huge increase in footfall as the percentage change from the baseline moved from -22.15% in CL-2021 to +9.3% in PL-Two-2021.

During CL-2021, in line with a significant decline in mobility, there was a substantial fall in the actual cases of RTD, RTI-Grievous and RTI-Minor by 64.45%, 78.53%, and 68.58%, respectively, when compared with the counterfactual. Moreover, the gravitation of actual cases of RTI-Grievous was singularly higher than RTD and RTI-Minor in CL-2021. The same trend of differential impact on RTI-Grievous was noticed during PL-One-2021 with the drop of 46.52%, 53.19%, 34.89% in the corresponding categories of RTC outcomes. Furthermore, it is noted that the overall impact was felt profoundly more in CL-2021 than in PL-One-2021. Sustained exceptional effect on RTI-Grievous was observed during PL-Two-2021 when the observed contractions were 44.68, 56.41%, and 32.41% with respect to RTD, RTI-Grievous and RTI-Minor, respectively. Perhaps, this can be attributed to the fact that many grievously injured victims of RTC during CL and PLs phases of the second wave chose not to visit the hospital when the severity and fatality of COVID -19 infection was high and preferred to lodge

complaints of the injury as minor as shown in Table 4. Thus, it is noted that in all the phases of lockdowns in the second wave, attendant factors influence the outcomes of road traffic crashes.

## Comparative analysis between the first and second wave

It is important to know whether the Government-enforced lockdown orders were synchronous with the rise and fall of COVID-19 infection. From (Fig 2), it is evident that the restrictive orders of CL-2020 and PL-2020 in the first wave were implemented well in advance before the pandemic could pose a serious challenge. It is probable that the pandemic's effect in the first wave was less severe because of these interventions. However, in the second wave, the Government-mandated orders such as CL-2021, PL-One-2021 and PL-Two-2021 were coterminous with the rise and fall of infection and fatalities due to the coronavirus.

As we have noted before, the fall in mobility during the first wave was unprecedented, which surprisingly did not replicate itself during the second wave even though there was a meteoric rise in the COVID-19 infection and fatality rate. In their study of the factors that impact RTD/RTI, the authors first compare the CL periods between the two waves. Second, they compare the PL phases of the two waves, followed by a comparison of post-lockdown phases of the two waves.

In the first comparison between two mobility distribution of CL -2020 in the first wave and CL-2021 in the second wave, low to medium effect size was noticed. Similarly, in the second comparison of the mobility distribution in PL-2020 and PL-Two-2021, large effect size was observed.

In the comparative periods of CL-2020 and CL-2021, small to medium effect size in mobility increase was strongly corelated to the marginal drop in declining percentages in RTD, RTI-Grievous, and RTI- Minor. The registered counterfactual decline in percentages in CL-2020 and CL-2021 were 78.73 and 64.45 for RTD; 83.77 vs 78.53 for RTI- Grievous and 80.13 vs 68.58 for RTI- Minor respectively, which is in conformity with the linear relationship between mobility and RTD/RTI as shown in Tables 4 and 5.

The RTD/RTI trends in the PL phases in the first wave and the second wave were on unexpected lines. The established relationship between mobility and RTD/RTI does not hold good as other attendant factors become the key predictors. There are two PL phases in the second wave. The authors compare PL-2020 in the first wave with PL-Two-2021 in the second wave. This is a more apt period for comparison than PL-One-2021 as PL-Two-2021 was enforced after the complete lockdown, just like the PL-2020 of the first wave.

In the comparison of PL-2020 with PL-Two-2021, we notice a reversed trend as RTD/RTI decreases with the increasing mobility of reported large effect size, and we also find that there is a differential impact to the outcomes of the road traffic crashes. We can see in Table 4 that the more substantial decline in the percentage of actual as against the counterfactual witnessed in RTI- Grievous in PL-2020 and PL-Two-2021 were 56.56% and 56.41%, whereas the corresponding changes in RTD were 44.83% and 44.68%; and in RTI- Minor the values were 49.33% and 32.41% respectively. In the second wave, the Government relaxed the restrictions especially in the PL-Two 2021; even though the pandemic continued to be relentless in claiming lives and inflicting hardships upon the infected. During this period, the health infrastructure was overwhelmed, the efforts and resources of various stakeholders in traffic safety were either diverted to attending to pandemic-related issues, or the personnel became severely infected or suffered from fear and apprehension of contracting the infection. It appears that, because of these circumstances, there was a sharper decline in the incidence of cases of RTI--Grievous when compared with the same spell in the first wave, pointing towards the role of attendant factors in the drop.

In the post-lockdown phases, it can be expected that when the restrictions are completely lifted, the incidence levels of RTD/RTI will bounce back to normal. The predicted and actual observed values of RTD/RTI would be the same as observed during the pre-COVID-19 period, that is, before the onset of the first lockdown on March 23, 2020. In Post-L-2020 of the first wave, once most restrictions were lifted, only marginal changes of -8.55% and -1.07% was noticed on RTD and RTI and a moderate effect on RTI- Grievous with a 13.6% drop. However, during Post-L-2021 in the second wave, there was a marked decline, especially in the case of RTI- Grievous, which plummeted by 87.24%. The RTD and RTI- Minor too had a significant decline of 50.07% and 16.79%, respectively (Figs 4–6 and Table 4). When the Post-L-2020 and Post-L-2021 periods were compared, the authors noticed a substantial increase in mobility in the latter phase, which reported medium and large effect sizes. However, there was a marked decline in the registration of road traffic injury cases. The decline in percentage with respect to RTI-Grievous during Post-L-2020 and Post-L-2021 were 13.6% and 87.24% respectively, and RTI- Minor was 1.07% and 16.79%, respectively.

The Post-L-2021 phase in the course of the second wave was markedly different in many ways. This phase was marked by relaxations in the restrictions announced by the Government, which enabled higher mobility in different spheres of activities in the state, though the pandemic was still present with moderate levels of spread of infection and fatalities. Thus, during the post lockdown phases in both the waves, especially in the second wave, the attendant factors outweighed the mobility or traffic flow in predicting road traffic injuries.

These anomalous relationships noticed above in the comparison of PL and Post-L periods of wave one and wave two bring to fore, in addition to mobility, the importance of attendant factors.

## Comparative analysis across different outcomes of RTC- namely RTD, RTI-Grievous and RTI-Minor during the first and second wave

The outcomes of a road traffic crash, namely RTD, RTI-Grievous and RTI-Minor, ought to have been impacted uniformly during the various phases of lockdown during both the pandemic waves in 2020 and 2021. The expectation was that all the three graphs would have similar patterns. While the RTD and RTI- Minor exhibited almost similar trends during the various phases of this study, RTI-Grievous seems to have been influenced differently, by some additional factors than just change in traffic flow. For instance, during CL-2021 in the second wave, RTD and RTI-Minor reported reductions of 64.45% and 68.58% as opposed to RTI--Grievous that reported 78.53%. The most marked difference was seen in the post-lockdown phase when a momentous decline of 87.24% in RTI- Grievous was recorded against a decline of 16.79% of RTI-Minor.

As stated earlier, one possible explanation is that since RTI-Grievous injuries require hospitalization for more than two weeks, many victims were not hospitalized for such a prolonged duration as the second wave of COVID-19 completely overwhelmed the health infrastructure and other allied facilities, including police and legal services. Moreover, the victims might also have refrained from visiting hospitals for medical treatment apprehending the likely infection of corona-virus in the environment of overcrowded health facilities with COVID-19 patients. In view of the above circumstances, the cases might have perhaps been registered as RTI-Minor, obviating the need for the victims of RTC to go to the hospital while being documented for claiming insurance. Thus, there is a possibility that there was no under-reporting of RTC, though there was a minimization of the offences from grievous to minor injury. The attendant factors outweighed the mobility factor as seen in the post-lockdown phase in 2021 when there was increased mobility; however, there was a decline in RTI-Grievous cases when compared

with the counterfactual, running contrary to the expectation of an increasing trend. These facilities were not availed by the victims of RTI- Minor.

Among the three outcomes of RTC, the differential impact on RTI-Grievous was felt only during the second wave of the pandemic in 2021. As stated earlier, the second wave of the pandemic was much severe and devastating when compared to the first one. The victims were unable to report and approach agencies like the police, fire and rescue services, ambulance, and legal resources as these were more focused and tasked to attend to corona-infected people. These are perhaps the reasons for the contra-trends noticed in RTI-Grievous during the phases in the second wave.

## Comparative analysis between Tamil Nadu and Chennai

The authors performed an individual analysis of TN state (130,060 $Km^2$ area; 72.1 million population) and Chennai city (175 $Km^2$ area; 8.65 million population) data to see if the road traffic crash trends and their relationship with other factors were the same for both regions. Injuries of all categories decreased in both regions, but some interesting trends were observed. In both regions, the drop in mobility was at a similar level in most spatial domains; however, during PL-One-2021 and PL-Two-2021 of the second wave, TN experienced an increase in mobility in grocery and pharmacy. Chennai City, in contrast, did not experience such an increase as shown in Tables 1 and 2. This may be due to a larger police presence in Chennai city compared to a leaner police presence in TN, thus leading to better enforceability of restrictive orders in Chennai city than in TN overall.

The RTD/RTI trends for TN and Chennai are not exactly similar during all lockdown phases, and there are some differences. All the three categories of road traffic crashes for TN and Chennai experienced a similar impact during CL-2020 and PL-2020 in the first wave. However, the trends in RTI- Minor cases during the various phases of lockdown in the second wave in Chennai city were largely different from the trends observed in TN. There was a sharp decline in RTI- Minor by 68.58% (cliff delta of -1) during CL-2021 of the second wave in TN as against a modest fall of 58.87% (cliff delta of -1) during the corresponding phase in Chennai city as shown in Tables 4 and 6. This is quite an anomalous trend as there was a steeper fall in mobility percentage in most sectors of activity in Chennai city than in TN during the relevant period. There were reverse trends in registration of RTI- Minor cases during the other phases, especially in PL-One-2021 and PL-Two-2021 of the second wave in TN and Chennai city. TN also reported a decline in PL-One-2021 and PL–Two-2021 by 34.89% (cliff delta of -1) and 32.41% (cliff delta of -1), respectively, while the corresponding drop in Chennai City was 33.63% and 39.22% as shown in Table 7 indicating that there was no significant difference between Chennai and TN in minor injury accident cases.

With regard to the registration of RTI-Grievous cases during the second wave, the impact was more substantial in TN than in Chennai city. In view of overall reduced mobility during lockdown phases, both TN and Chennai city exhibited decreased levels of RTI-Grievous incidences. However, TN had more mobility than Chennai city in most spheres of activity during CL and PL phases of the second wave. The reduction trends in RTI- Grievous also exhibited similar patterns in TN and Chennai city, with a more substantial reduction in TN in all phases in the first wave and the second wave when compared with a lower reduction in Chennai city. The figures of RTI-Grievous during CL-2021 in TN and Chennai city are -78.53%, and– 53.32%, and in PL-One-2021 are -53.19% and -29.87%; and in PL-Two-2021–56.41% and -40.39% respectively as shown in Tables 4 and 6. There are 7.183 Hospitals for 1,00,000 people in Chennai City as against 3.133 for Tamil Nadu [26–28]. Similarly, the ratio for the number of police officers for every 1,00,000 people in Chennai is 410, and Tamil Nadu is 129. This

**Table 6. Showing the actual and predicted daily counts of RTD/RTIs in all phases during both waves for Chennai.**

| PERIOD | Metric/RTC | RTD | RTI—Grievous | RTI—Minor |
|---|---|---|---|---|
| CL 2020 | Actual | 4 | 13.67 | 12 |
| | Predicted Mean | 24.49 | 64.10 | 77.89 |
| | Percentage Change | -83.67 | -78.68 | -84.59 |
| PL 2020 | Actual | 11.17 | 32.50 | 30.67 |
| | Predicted Mean | 23.09 | 61.58 | 75.89 |
| | Percentage Change | -51.63 | -47.22 | -59.59 |
| Post–L 2020 | Actual | 19.71 | 65.71 | 69.43 |
| | Predicted Mean | 21.79 | 54.49 | 68.47 |
| | Percentage Change | -9.55 | 20.60 | 1.40 |
| PL–One 2021 | Actual | 12 | 31.33 | 52.33 |
| | Predicted Mean | 21.66 | 44.68 | 78.85 |
| | Percentage Change | -44.60 | -29.87 | -33.63 |
| CL 2021 | Actual | 9 | 22.67 | 32 |
| | Predicted Mean | 22.80 | 48.56 | 77.8 |
| | Percentage Change | -60.52 | -53.32 | -58.87 |
| PL–Two 2021 | Actual | 7.67 | 26.17 | 44.5 |
| | Predicted Mean | 23.53 | 43.89 | 73.21 |
| | Percentage Change | -67.42 | -40.39 | -39.22 |

once again highlights the role of attendant factors and, more importantly, victims' easier access to police and robust health infrastructure in Chennai City than in TN.

A similar analysis has been done for RTD. The percentages of reduction of registration of RTD in TN and Chennai during CL-2020 are -78.73 (cliff delta of -0.9) and -83.67 (cliff delta of -1), during CL-2021 are 64.45 (cliff delta of -1) and 60.52 (cliff delta of -1), during PL-One 46.52 (cliff delta of -1) and 44.60 (cliff delta of -1), and during PL-Two are 44.68 (cliff delta of -0.94) and 67.42 (cliff delta of -1). It is seen that there is very little difference between TN and Chennai. It is observed that in the comparative study of the impact of lockdowns in TN and

**Table 7. Percentage changes in mean and effect size (Cliff s Delta) of TN and Chennai.**

| Period/Place | | Tamil Nadu | | | Chennai | | |
|---|---|---|---|---|---|---|---|
| | Metric/RTC | RTD | RTI-G | RTI-M | RTD | RTI-G | RTI-M |
| CL 2020 | %Change | -78.73 | -83.77 | -80.13 | -83.67 | -78.68 | -84.59 |
| | Cliff D | -0.90 | -1 | -1 | -1 | -1 | -1 |
| | Cliff CI | -1, -0.8 | -1, -1 | -1, -1 | -1, -1 | -1, -1 | -1, -1 |
| PL 2020 | %Change | -44.83 | -56.56 | -49.33 | -51.63 | -47.22 | -59.59 |
| | Cliff D | -0.904 | -1 | -1 | -1 | -1 | -0.89 |
| | Cliff CI | -0.79,-0.98 | -1, -1 | -1, -1 | -1, -1 | -1, -1 | -1, -0.78 |
| PL–One 2021 | %Change | -46.52 | -53.19 | -34.89 | -44.6 | -29.87 | -33.63 |
| | Cliff D | -1 | -1 | -1 | -1 | -1 | -1 |
| | Cliff CI | -1, -0.99 | -1, -1 | -1, -1 | -1, -1 | -1, -1 | -1, -1 |
| CL—2021 | %Change | -64.45 | -78.53 | -68.58 | -60.52 | -53.32 | -58.87 |
| | Cliff D | -1 | -1 | -1 | -1 | -1 | -1 |
| | Cliff CI | -1, -1 | -1, -1 | -1, -1 | -1, -1 | -1, -1 | -1, -1 |
| PL–Two 2021 | %Change | -44.68 | -56.41 | -32.41 | -67.42 | -40.39 | -39.22 |
| | Cliff D | -0.94 | -1 | -0.84 | -1 | -0.72 | -0.67 |
| | Cliff CI | -1, -0.88 | -1, -1 | -0.98,-0.69 | -1, -1 | -0.82,-0.62 | -0.74,-0.6 |

Chennai city, the impact in RTD was the same, whereas, in RTI-Grievous, Chennai city had a lesser impact.

The overall differential impact may perhaps be attributed to the overall better health infrastructure and access to police in Chennai city than in TN. This enabling attendant factor prompted victims to seek medical treatment more frequently in Chennai. The apprehension of the road traffic crash victims to not seek medical treatment on account of infection in the overcrowded hospitals with COVID-19 patients, though partly true, was not applicable to Chennai city. The availability of hospitals, especially in the private sector in Chennai city, ensured treatment only to patients who tested negative for the coronavirus, was a huge enabling factor in road traffic victims in Chennai city seeking help in comparison to victims in TN.

## Role of attendant factors and mobility

This study spans across the two waves of COVID-19 pandemic and the varying degrees of restrictions imposed by the Government in an attempt to contain the spread of the virus. These periods of restrictions due to the pandemic induced lockdown directly impacted mobility and thereby present a timely opportunity to explore the relationships between RTD/RTI and a number of other concomitant factors, such as victims' access to relief and rehabilitation centres, the capability of health and other allied infrastructure, and responsiveness of police and legal services. These accompanying factors often account over and beyond the rise and fall in mobility.

Under usual circumstances, an increase in mobility or higher traffic flow results in an increase in RTC. Hence mobility as a factor could reasonably help us predict the rate of RTD/ RTI during normal times. This factor was used to predict the rate of RTD/RTI during both the first and second waves of the pandemic. It should be noted here that the first wave was not as severe as the second wave. However, the results from the second wave of the pandemic in TN signals toward an anomalous relationship between mobility and RTD/RTI during the different phases of the lockdown. The second wave witnessed an inverse relationship between mobility and the rate of RTD/RTI, and could lead to consideration of attendant factors, such as transportation and medical treatment of road traffic crash victims, minimal access to other emergency services, and the police assuming a greater significance than the magnitude of traffic flow. The decline in RTI-Grievous during the second wave in the periods of PL-One-2021, PL-Two-2021 and Post-L-2021 was substantially higher than RTI-Minor as the latter category did not require hospitalization of victims. Out of the three outcomes, victims of RTI-Minor require the least support from hospitals in the form of first aid. For victims of RTD, the hospitals only have to conduct the post-mortem of the deceased victim. The victims of RTI-Grievous require the maximum care and attention from the hospitals for relief, medical treatment, and rehabilitation. Moreover, the victims of RTI-Grievous usually require a few weeks of hospitalization, and the injury also needs to be legally classified as 'Grievous'.

The CL and PL phases of the first wave witnessed direct relationship between mobility and incidence of RTD/RTI, and seemed to point towards a diminished or minimal role of attendant factors. In addition to that, the pandemic infection and fatality rate during the first wave was not as severe as wave two. Furthermore, the pandemic-related strain on hospital infrastructure was present in both the waves but the strain during the first wave was on a much lower scale than during the second wave. However, in wave one, people were exposed to this pandemic for the first time in their lives, and it instilled fear, apprehension, and worry, eventually resulting in a historic fall in mobility, as people implicitly obeyed the stay-at-home orders.

Contrary to the belief, when the mobility increased during PL and Post-L in wave two in 2021, there was a drop in RTD/RTI. Interestingly, the steepest decline was in the recorded levels of RTI-Grievous, and a drop in registered cases of RTD and RTI-Minor were less

significant. When the people encountered the more devastating wave two of the pandemic, crowded environment of health facilities with COVID-patients deterred even victims of road traffic crashes to not access these centres for remedial relief. Furthermore, every resource of the hospital infrastructure was committed to cater to the pandemic workload.

These results indicate the interaction between mobility and attendant factors in influencing the occurrence of RTC/RTDs. During the less severe first wave when the attendant factors were almost in a similar state prior to pandemic (health infrastructure and police needs were not overwhelming though little higher than pre-covid levels), the changes in mobility caused changes in the occurrence of RTC/RTDs. But during the more severe second wave the demand for attendant factors was overwhelmingly high. This difference in attendant factors between two waves helped bring out the influence of attendant factors. And the magnitude of the effect of attendant factors is brought out by the percentage changes in RTC/RTD levels in periods where we observe an inverse relationship, thus highlighting the influence of attendant factors.

Chennai City is better placed in terms of a higher police to people ratio than the rest of Tamil Nadu. Adding on to that, Chennai city boasts of superior hospital infrastructure, substantially augmented health and allied services, including ambulance and trauma care centres than the rest of Tamil Nadu. After adjusting for mobility in wave one, TN and Chennai had very similar trends in RTD/RTI. However, in wave two, the impact was differential. The reduction of RTI-Grievous in TN was significantly more than in Chennai City. This highlights the robustness of the first responders; namely police, ambulance, trauma care centres, and hospitals in Chennai and stresses the importance of attendant factors in determining the recorded levels of RTI-Grievous cases.

## Traffic flow management during pandemic

The traffic flow management did witness substantial altered changes in terms of the kind of vehicles and timing of their movement during the complete and partial lockdowns. The crucial changes during complete lockdown were that private vehicles, including two-wheelers, were not allowed to ply and vehicles connected with essential supply and services were permitted to operate. Generally, these vehicles fall under the category of heavy and medium vehicles instead of privately owned light motor vehicles such as cars, SUVs and two-wheelers. During this period, the mobility of people and vehicles was minimal.

When the relaxations were introduced in partial lockdown, the number of vehicles did not increase significantly, as evidenced by the Google Mobility Community Report data. Furthermore, during the partial lockdown period, only certain limited kinds of vehicles were permitted to operate and that too in specific restricted time windows. Even in these periods, public transportation such as buses, taxis, and other operators had not opened up. Because of these changes in density, the timing of vehicular traffic, type of vehicles and pedestrian traffic, the resources connected with law enforcement dealing with traffic were reorganized and also shared their commitment with pandemic related work.

In Chennai city, the police, including officers of the traffic department, enforced the lockdown measures more stringently than in rest of Tamil Nadu in booking cases and impounded the vehicles that violated the restrictive orders. However, it did not impact the incidence of road traffic crashes [29].

There have been instances of anomalous behaviour of road users resulting in more road traffic crashes in some provinces in other countries during the restrictive phases of stay-at-home orders. Our research findings did not find any such increase, perhaps due to the absence of errant behaviour included over speeding, drunk driving, not wearing seat belts in Tamil Nadu during the restrictive periods of the pandemic.

## Limitations of the study

The study is entirely based on the data provided by crimes registered as FIR for road crashes from before and after the onset of the pandemic. Unlike violent crimes or property offences, the degree of underreporting in RTC is lower because the victims of road crashes need to produce First Information Report (FIR) as documentation to make an insurance claim. Insurance coverage is a mandatory requirement for all vehicles, but the insurance purchase is not compulsory or obligatory for properties and life. Therefore, underreporting of road crashes might be of less concern. Nevertheless, there are instances of under-reporting in the Indian context [30]. The victim may not report the crime, or the incidence of road crashes for various reasons. A few people feel that it is not necessary to report crashes to the police. For example, in "hit and run" cases, the details of the alleged vehicle causing the crash are not known. Usually, the victim of a road crash gets compensation from the company that provides the insurance cover for the vehicle. In the absence of the details of the vehicle and the corresponding insurance company, the victim has little incentive to file a police report. Therefore, the victim is less likely to report such crashes [31].

The under-reporting for such reasons does not impact the study as the causal inference drawn based on the forecast is constructed upon past observations where this inadequacy of under-reporting was present. As this factor has been present prior, during, and post-intervention, it does not pose a problem.

Google Mobility Community Report is only an aggregate indicator of the traffic flow, which is not true during PL phases. During the course of the day, there is a heavy density of traffic in the permitted hours and a very lean flow of vehicles during the prohibited hours. Drawing conclusions based on aggregated indices of mobility to draw inferences on the relationship between mobility and incidence of RTC is not accurate but only an approximation.

Forecasting long into the future (nearly 547 time stamps) may not yield accurate forecasts.

## Policy implications

The research findings had clearly identified the serious inadequacies in various relief and rehabilitation centres to the victims of road traffic crash when the state was overwhelmed by the pandemic. This also highlights the positive response of the same institutions when the pandemic was less severe. The drastic decline in RTI-Grievous during all the phases of the second wave maybe because some of these cases would have ended in fatalities, or the cases were registered as RTI-Minor in view of pre-occupied health infrastructure that was dealing with the pandemic. The results of this study highlight that such loss of lives thus could have been averted, and it calls for the attention of the policymakers in taking appropriate measures in strengthening the facilities during the times of crisis like the pandemic.

## Conclusion

The direct relationship between traffic flow and RTD/RTI-Grievous and RTI-Minor, as noticed during the usual circumstances, was not observed during all phases of lockdowns in the first and second waves of COVID-19. There was an unprecedented fall in the cases of RTD/RTI in TN during CL-2020 and PL-2020 of wave one. This phenomenal drop was not noticed even during any other phase of lockdown in the second wave of COVID-19 in TN, when it took a heavier toll on lives and caused an upsurge of infection cases. The drastically reduced mobility paved the way for a reduction in RTD/RTI in TN only in the first wave, aligning with the already established relationship between traffic flow and road traffic crashes. As the second wave was more severe and devastating than the first one, there were enormous challenges for the victims of road traffic crashes to access the major relief and rehabilitation

centres of police, legal, hospital, and other allied services. These accompanying factors were more crucial than mobility in predicting the incidence of cases of road traffic injuries. In view of these changed circumstances, instead of the well-established linear relationship between mobility and RTI, during wave two, contrary trends of the decline of RTI- Grievous were noticed when there was increased mobility. The government policy of enforcement of stay-at-home orders in varying degrees facilitated the containment of pandemic and also, importantly, created certain positive externalities such as reducing road crashes and a cleaner environment. However, when the important responders of the road traffic victims, such as police, hospital, trauma care centres, were overwhelmed as witnessed during the second wave, increased registration of RTI-Minor with a corresponding steep decline in the registration of RTI-Grievous pointed to the inability of the victims to seek adequate treatment. Therefore the important takeaway of the research is the need to focus on the road traffic crash victims and public service delivery systems stakeholders such as police, hospitals, ambulance service, legal service, and better reporting and seeking channels of victims of road traffic crashes for treatment,

While Chennai and TN had similar trends during the first wave, improved medical and supporting infrastructure, a higher concentration of police officers, easier access to relief centres in the urban city of Chennai did have a positive impact in processing the victims of road traffic crashes as noticed in higher registration of road traffic injuries during the second wave.

## Supporting information

**S1 Data. Accident data–Tamil Nadu.**
(CSV)

**S2 Data. Accident data–Chennai.**
(CSV)

**S3 Data. Mobility data–Tamil Nadu.**
(CSV)

**S4 Data. Mobility data–Chennai.**
(CSV)

## Acknowledgments

The authors are extremely grateful to the State Crime Records Bureau, Tamil Nadu Police Department, India and the State Police Master Control Room, Chennai, India for providing the data required for this research.

Further, the authors wish to thank Mr. Raghuraman, Inspector of Police, and Mrs Shanthi Sub- Inspector of Police, Technical Services, Tamil Nadu Police, for their assistance in the collection and preparation of data. We thank Ms. Pooja Srivatsan for language editing and formatting.

The authors are thankful to the State Crime Records Bureau, Tamil Nadu Police Department, Chennai for providing data pertaining to RTCs in Tamil Nadu, and the National Crime Records Bureau's annual publication, Crime in India, for data on other states.

## Author Contributions

**Conceptualization:** Kandaswamy Paramasivan, Nandan Sudarsanam.

**Data curation:** Rahul Subburaj, Venkatesh Mohan Sharma.

**Formal analysis:** Kandaswamy Paramasivan, Rahul Subburaj.

**Funding acquisition:** Nandan Sudarsanam.

**Investigation:** Kandaswamy Paramasivan.

**Methodology:** Rahul Subburaj.

**Software:** Rahul Subburaj.

**Supervision:** Nandan Sudarsanam.

**Validation:** Rahul Subburaj.

**Visualization:** Venkatesh Mohan Sharma.

**Writing – original draft:** Kandaswamy Paramasivan.

**Writing – review & editing:** Kandaswamy Paramasivan, Rahul Subburaj, Venkatesh Mohan Sharma, Nandan Sudarsanam.

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
