## [Decision Letter · Decision Letter 0]

2 Mar 2022

PONE-D-21-40969Relationship between mobility and road traffic injuries during COVID-19 pandemic – the role of attendant factorsPLOS ONE

Dear Dr. Paramasivan,

Thank you for submitting your manuscript to PLOS ONE. After careful consideration, we feel that it has merit but does not fully meet PLOS ONE’s publication criteria as it currently stands. Therefore, we invite you to submit a revised version of the manuscript that addresses the points raised during the review process.==

We look forward to receiving your revised manuscript.

Kind regards,

Jian Wang

Academic Editor

PLOS ONE

Journal Requirements:

2. Please note that PLOS ONE has specific guidelines on code sharing for submissions in which author-generated code underpins the findings in the manuscript. In these cases, all author-generated code must be made available without restrictions upon publication of the work. Please review our guidelines at https://journals.plos.org/plosone/s/materials-and-software-sharing#loc-sharing-code and ensure that your code is shared in a way that follows best practice and facilitates reproducibility and reuse. Code may be shared by providing a URL within the Methods section to a code repository or it may be uploaded as a supplemental file

“The authors also acknowledge the support provided by the Robert Bosch Centre for Data Science and Artificial Intelligence (RBCDSAI) and Indian Institute of Technology Madras, India (SB20210605MSRBCX008658).”

“NO- The funders had no role in study design, data collection and analysis, decision to publish or preparation of the manuscript”

“NO- The funders had no role in study design, data collection and analysis, decision to publish or preparation of the manuscript”

“NO authors have competing interests”

Reviewers' comments:

Reviewer's Responses to Questions

**Comments to the Author**

1. Is the manuscript technically sound, and do the data support the conclusions?

Reviewer #1: Yes

Reviewer #2: Yes

2. Has the statistical analysis been performed appropriately and rigorously? 

Reviewer #1: Yes

Reviewer #2: Yes

3. Have the authors made all data underlying the findings in their manuscript fully available?

Reviewer #1: Yes

Reviewer #2: Yes

4. Is the manuscript presented in an intelligible fashion and written in standard English?

Reviewer #1: Yes

Reviewer #2: No

5. Review Comments to the Author

Reviewer #1: This paper uses the blockade background caused by the COVID-19 coronavirus disease epidemic to study road traffic safety. This is a novel and interesting work, and the article has clear ideas. However, the following problems still exist:

1、 The abstract and introduction of this paper do not refine the research and innovation of traffic safety under the background of pandemic.

2、 The comparison of the impact on RTC, RTI and RTD in the first and second wave blockades caused by the pandemic needs a sufficient amount of data to support. How to deal with and analyze a large amount of data?

3、 The article should describe in more detail the parameters of the proposed prediction model (equation 4) and how to apply the prediction.

4、 The reason for the difference between the predicted time series and the actual time series is whether the blockade implemented by the government has limitations and other influencing factors are not considered.

5、 The training time and experimental time of the model reach 12 months and 4 months respectively, but the verification time is only 45 days. Whether the verification time is too little leads to the lack of persuasion of the model.

6、 This paper mainly expounds the relationship between traffic flow and various accompanying factors in the context of disease pandemic, but the summarized relationship between them does not explain the actual traffic flow management and what measures to take in the context of pandemic.

Reviewer #2: This paper investigates the imapct of attendant factors on relationship between mobility and road traffic injuries during COVID-19 pandemic. overal, the paper is well-writen, and results are meaingful. However, there is a problem as follow:

As explained in the title, this paper focuses on the the role of attendant factors (road traffic victims' access to trauma centres, the robustness of health infrastructure, and the responsiveness of police and emergency services). However in the section "3. Results" and "4. Discussion", results of the role of attendant factors are not displayed prominently although lots of results and comparisons are provided. It is suggested to reorganize the two sections to highlight results related to the role of attendant factors.

6. PLOS authors have the option to publish the peer review history of their article (what does this mean?). If published, this will include your full peer review and any attached files.

Reviewer #1: **Yes: **Chao Sun

Reviewer #2: No

---

## [Author Response · Author response to Decision Letter 0]

21 Mar 2022

RESPONSE TO REVIEWERS

Reviewer #1: This paper uses the blockade background caused by the COVID-19 coronavirus disease epidemic to study road traffic safety. This is a novel and interesting work, and the article has clear ideas. However, the following problems still exist:

1. The abstract and introduction of this paper do not refine the research and innovation of traffic safety under the background of pandemics.

We had made an effort to present the research on traffic safety during regular times and the pandemic. Specifically, paragraph three of the introduction in the previous draft is dedicated to positioning our work viz-a-viz the literature on road safety during the pandemic. However, based on the reviewer's comment, we felt it might be missed in its current state. We have now made three sub-sections under introduction providing the background, literature review covering both pre-pandemic and pandemic, and finally focusing on the innovation in the present research. 

2. The comparison of the impact on RTC, RTI and RTD in the first and second wave blockades caused by the pandemic needs a sufficient amount of data to support. How to deal with and analyze a large amount of data?

 The data in the study is the daily count of road traffic crashes for eleven years from 1st January 2010 to 30th September 2021. Each time series contains 4291 data points. Moreover, the data used in the study corresponds to two geographic locations, making the data large and complex. Large data usually poses problems in two aspects: the storage and cleaning, and the modelling part, where we try to draw inferences from the data. 

The first aspect was not a significant issue in this study as the data is a univariate time series (we have three such time series) with one value for each time step which is easy to store as compared to data with multiple features for a single data point (tabular data) or images. All of the time steps had non-empty values at each time step, so not demanding the need for data cleaning. 

In the modelling phase, the DeepAR model used in the study belongs to a class of deep learning models that are highly suitable for learning from large time-series data, unlike commonly used models like ARIMA and exponential smoothing, which are not so robust in large data settings and saturate in performance. While deep learning models take more extended periods to train, we used GPU based hardware architecture to train our DeepAR model in this study, which drastically reduces the training time. Therefore, the right choice of the model and the hardware can be used to overcome the issues in modelling with a large dataset. 

A brief description of the points mentioned here is added in the paper titled "Method ".

3. The article should describe in more detail the parameters of the proposed prediction model (equation 4) and how to apply the prediction.

The section entitled "Prediction" is revised to include the detailed process behind generating a forecast. After the revision, equation (4) mentioned in the paper is made clearer and more comprehensive, summarising the steps involved using mathematical notation. The hyper-parameters of the best model are mentioned in the section now entitled "Parameters and forecast accuracy of model". The section also includes the lag terms present in the best model used to generate the forecasts. 

4. The reason for the difference between the predicted time series and the actual time series is whether the blockade implemented by the government has limitations, and other influencing factors are not considered.

 A direct relationship between mobility and RTC/RTD is well established in the literature and referenced in our paper (When mobility increases, the incidence of RTC/RTD increases). The COVID-19 induced blockade restricted the mobility of people, which is evident from the google mobility data used in the study. In wave one, when the severity of the infection was mild, the downward trend in RTD/RTI was in alignment with the mobility. However, in wave two, even after adjusting for mobility, the actual daily count of RTI-Grievous was substantially lower than predicted. This is primarily attributable to the attendant factors such as overwhelmed health infrastructure and apprehension and disinclination of the victims to seek relief at hospitals. In other words, mobility was a reasonable predictor of RTD/RTI; however, in exceptional circumstances such as the pandemic in wave two, the attendant factors influenced more than mobility.

The blockade's implementation was more effectively done in Chennai city than the rest of Tamil Nadu because of a higher ratio of police officers to people in Chennai City. Similarly, the people's compliance to the restrictive orders also varied across time. However, these variations have clearly manifested into appropriate changes in mobility which forms the basis of the study. The other influencing factors of the pandemic induced blockades have been effectively captured through mobility.

5. The training time and experimental time of the model reach 12 months and 4 months respectively, but the verification time is only 45 days. Whether the verification time is too little leads to the lack of persuasion of the model

The DeepAR model is trained using data from 1st January 2010 to 31st December 2019 (10 years). The validation period of the model is from 1st January 2020 to 22nd March 2020 (82 days), and the prediction period is from 23rd March 2020 to 30th September 2021 (557 days) (The periods are mentioned in the section entitled "Data"). Increasing the validation period has two issues: first, reducing the data available for training the model leads to underfitting. Second, increasing the validation size does not allow the model to learn the recent/short term relationship. Hence, a validation period of 45 days is used in the study. As is standard practice, the validation output is also unseen by the learning model. 

6、 This paper mainly expounds on the relationship between traffic flow and various accompanying factors in the context of disease pandemic, but the summarized relationship between them does not explain the actual traffic flow management and what measures to take in the context of pandemic.

We have explained in the section under "policy implications" the measures to be taken in the context of pandemics relating to road traffic incidents. However, considering the reviewer's remark, in this revision, we have included the summarized relationship in the CONCLUSION section as well. Further, we have also added another sub-section under "Discussion" entitled "Traffic flow management during the pandemic."

Reviewer# 2

This paper investigates the impact of attendant factors on the relationship between mobility and road traffic injuries during the COVID-19 pandemic. Overall, the paper is well-written, and the results are meaningful. However, there is a problem as follow:

As explained in the title, this paper focuses on the role of attendant factors (road traffic victims' access to trauma centres, the robustness of health infrastructure, and the responsiveness of police and emergency services). However, in the sections "3. Results" and "4. Discussion", results of the role of attendant factors are not displayed prominently, although lots of results and comparisons are provided. It is suggested to reorganize the two sections to highlight results related to the role of attendant factors.

We thank the reviewer for pointing out the need to highlight the results of the role of attendant factor prominently. Besides revising the " Discussion " appropriately, we have specifically added an exclusive sub-section entitled the role of attendant factors and mobility in the main DISCUSSION section, besides revising the “Discussion” appropriately. Further, we have added a few lines in the conclusion section to highlight the key takeaways of the research in dealing with these attendant factors during the crisis and future.

---

## [Decision Letter · Decision Letter 1]

25 Apr 2022

Relationship between mobility and road traffic injuries during COVID-19 pandemic – the role of attendant factors

PONE-D-21-40969R1

Dear Dr. Paramasivan,

We’re pleased to inform you that your manuscript has been judged scientifically suitable for publication and will be formally accepted for publication once it meets all outstanding technical requirements.

Kind regards,

Jian Wang

Academic Editor

PLOS ONE

Additional Editor Comments (optional):

Reviewers' comments:

Reviewer's Responses to Questions

**Comments to the Author**

1. If the authors have adequately addressed your comments raised in a previous round of review and you feel that this manuscript is now acceptable for publication, you may indicate that here to bypass the “Comments to the Author” section, enter your conflict of interest statement in the “Confidential to Editor” section, and submit your "Accept" recommendation.

Reviewer #1: All comments have been addressed

Reviewer #2: All comments have been addressed

2. Is the manuscript technically sound, and do the data support the conclusions?

Reviewer #1: Yes

Reviewer #2: Yes

3. Has the statistical analysis been performed appropriately and rigorously? 

Reviewer #1: Yes

Reviewer #2: Yes

4. Have the authors made all data underlying the findings in their manuscript fully available?

Reviewer #1: Yes

Reviewer #2: Yes

5. Is the manuscript presented in an intelligible fashion and written in standard English?

Reviewer #1: Yes

Reviewer #2: Yes

6. Review Comments to the Author

Reviewer #1: Thanks for the efforts of responding my comments. You have addressed my concerns to a satisfactory level.

Reviewer #2: The author has revised the paper according to my suggestion, now I am happy to recommend this paper for acceptance.

7. PLOS authors have the option to publish the peer review history of their article (what does this mean?). If published, this will include your full peer review and any attached files.

Reviewer #1: **Yes: **Chao Sun

Reviewer #2: No

---

## [Editor Report · Acceptance letter]

13 May 2022

PONE-D-21-40969R1 

Relationship between mobility and road traffic injuries during COVID-19 pandemic – The role of attendant factors 

Dear Dr. Paramasivan:

I'm pleased to inform you that your manuscript has been deemed suitable for publication in PLOS ONE. Congratulations! Your manuscript is now with our production department. 

Kind regards, 

on behalf of

Dr. Jian Wang 

Academic Editor

PLOS ONE